# Blood DNA methylation profiling identifies cathepsin Z dysregulation in pulmonary arterial hypertension

Anna Ulrich [1,21], Yukyee Wu [2,21], Harmen Draisma [1,3], John Wharton [2], Emilia M. Swietlik[4], Inês Cebola [3], Eleni Vasilaki[2], Zhanna Balkhiyarova[1,3,5], Marjo-Riitta Jarvelin [6,7,8,9], Juha Auvinen[7], Karl-Heinz Herzig [10,11], J. Gerry Coghlan[12], James Lordan[13], Colin Church[14], Luke S. Howard [2], Joanna Pepke-Zaba[15], Mark Toshner[4], Stephen J. Wort [2,16], David G. Kiely[17,18,19], Robin Condliffe[17,18], Allan Lawrie [2,17], Stefan Gräf [4,20], Nicholas W. Morrell [4], Martin R. Wilkins [2], Inga Prokopenko [1,22] & Christopher J. Rhodes [2,22] ✉

Pulmonary arterial hypertension (PAH) is characterised by pulmonary vascular remodelling causing premature death from right heart failure. Established DNA variants influence PAH risk, but susceptibility from epigenetic changes is unknown. We addressed this through epigenome-wide association study (EWAS), testing 865,848 CpG sites for association with PAH in 429 individuals with PAH and 1226 controls. Three loci, at Cathepsin Z (*CTSZ*, cg04917472*)*, Conserved oligomeric Golgi complex 6 (*COG6*, cg27396197*)*, and Zinc Finger Protein 678 (*ZNF678*, cg03144189*)*, reached epigenome-wide significance ($p < 10^{-7}$) and are hypermethylated in PAH, including in individuals with PAH at 1-year follow-up. Of 16 established PAH genes, only cg10976975 in *BMP10* shows hypermethylation in PAH. Hypermethylation at *CTSZ* is associated with decreased blood cathepsin Z mRNA levels. Knockdown of CTSZ expression in human pulmonary artery endothelial cells increases caspase-3/7 activity ($p < 10^{-4}$). DNA methylation profiles are altered in PAH, exemplified by the pulmonary endothelial function modifier *CTSZ*, encoding protease cathepsin Z.

Pulmonary arterial hypertension (PAH) is a rare and highly morbid disease. In affected individuals it leads to ~10% annual mortality due to right heart failure, driven by a high vascular resistance in the lungs[1]. In its idiopathic form, PAH mostly affects women with an incidence of 25 individuals per 1 million population in Western countries, and an annual incidence of 2 to 5 newly affected per million[2]. The underlying pathology, comprising vasoconstriction and pulmonary vascular remodelling, is not addressed adequately by currently available therapies, reflecting our limited understanding of the molecular drivers of PAH[3].

Mutations in bone morphogenetic protein receptor 2 (*BMPR2*) are found in approximately 80% of individuals with hereditary PAH

(~6% of PAH cases), and in up to 20% of individuals with idiopathic disease. Disease penetrance in carriers is estimated at 20%, meaning that a range of additional factors including genome-wide polygenic DNA variability, external and potentially epigenetic factors contribute to individual susceptibility. In PAH, there is also a discrepancy between sexes in penetrance, where *BMPR2* mutations (chromosome 2) lead to development of PAH in 12% of male but 42% of female carriers[4]. There is a clear need to disentangle the factors that determine PAH susceptibility and *BMPR2* mutation penetrance.

The role of epigenetic variation in PAH is intriguing, since *BMPR2* promoter methylation is elevated in PAH individuals compared to unaffected *BMPR2* mutation carriers[5], offering one mechanism for the

---

varying penetration of heritable PAH and a potential target for pharmacological intervention. Alterations of CpG methylation levels in rodent models of pulmonary hypertension (PH – which can include pulmonary venous hypertension) have been demonstrated and successfully targeted. DNA methyltransferase inhibition can rescue the hypoxia-driven gene methylation and consequent decrease in phosphatase *PTEN* expression leading to inhibition of pathologic proliferation, migration and promotion of apoptosis in rat pulmonary arterial smooth muscle cells (PASMC)[6]. In mice, hypoxia-induced PH was prevented by hypomethylation of a promoter-flanking region of the cell growth-inhibiting non-coding RNA *lncPINT*, induced by glucose-6-phosphate dehydrogenase inhibition[7].

The broader picture of genomic methylation in human PH remains largely unknown, despite a decade of successes of epigenome-wide association studies (EWAS) for more common phenotypes. Comparative analysis of whole epigenome-wide DNA methylation profiles of cultured patient-derived PASMCs from chronic thromboembolic PH (CTEPH) and control cell lines detected 6829 significantly differentially methylated probes (DMPs) between these two groups[8]. Among these, 4246 DMPs were hypermethylated, while 2583 DMPs were hypomethylated[8]. In pulmonary artery endothelial cells from individuals with idiopathic and heritable PAH and healthy controls, unsupervised hierarchical clustering identified 147 differentially methylated promoters, 46 of which code for proteins or microRNAs related to lipid metabolism[9].

DNA methylation (DNAme) is the most widely studied type of epigenetic mark. In humans, most DNAme comprises the addition of a methyl group to cytosine bases within cytosine-guanine (CpG) DNA sequences, usually concentrated in relatively high densities, referred to as CpG islands. Alterations in DNAme, which subsequently propagate through mitosis, can occur in response to both external environmental and internal stimuli, while some proportion of variation in DNAme appears to be random in nature[10]. DNAme and other epigenetic processes can affect gene expression, thereby modifying cellular phenotypes and contributing to a disease risk. The presence of a methyl group at a CpG island can hinder DNA recognition and transcription factor binding[11]; alternatively, the methyl group could recruit proteins that preferentially bind to methylated DNA and prevent access to the promoter site by transcription factors[12]. EWAS of complex diseases have successfully identified disease-relevant epigenetic marks from blood[13–15]. DNAme can be measured in peripheral blood

samples; in particular, through the analysis of white blood cells. While not pulmonary vascular tissue, inflammation plays an active role in PAH[16] and white blood cells provide an insight into epigenetic changes that might be shared with vascular tissue.

In this EWAS, we sought to determine the global epigenomic profile of PAH in a large, well-defined cohort of 429 individuals with PAH and 1206 controls using the Illumina EPIC array in peripheral blood samples from the UK National Cohort Study of Idiopathic and Heritable PAH (hereon referred to as the PAH Cohort Study, clinicaltrials.gov: NCT01907295, www.ipahcohort.com) and three external studies.

## Results

### EWAS dataset and array

We assayed DNA methylation profiles of peripheral blood of individuals with idiopathic, heritable, or drug-induced PAH (*n* = 454, 429 after exclusions/quality control) and controls (*n* = 1226) using the Illumina Infinium MethylationEPIC BeadChip Kit, which covers over 850,000 methylation sites (Methods). Additionally, DNA methylation was measured at a second timepoint for 341 of the 454 individuals with PAH. The CPACOR pipeline (from Lehne et al.[17]) was used for the processing on genome build GRCh37/hg19, quality control (QC) and normalisation of the raw methylation data (Fig. 1, Methods). Within the PAH Cohort Study, we implemented the internationally approved diagnostic criteria for idiopathic and heritable PAH, specifically, a raised mean pulmonary artery pressure (mPAP) ≥25 mmHg with pulmonary capillary wedge pressure (PCWP) ≤ 15 and pulmonary vascular resistance (PVR) > 3 Wood units at rest, and exclusion of known associated diseases[16]. In addition to our healthy control group (*n* = 106), we obtained raw whole blood methylation data from three external sources including the Parkinson's Progression Markers Initiative (PPMI, *n* = 266 after exclusions), the Alzheimer's Disease Neuroimaging Initiative (ADNI, *n* = 141) and the population-based Northern Finland Birth Cohort 1966 (NFBC1966, *n* = 694) through collaboration and these served as controls. We excluded all individuals over 80 years of age, all participants with Alzheimer's disease or unknown disease status (ADNI) and all participants with Parkinson's disease aged over 60 (PPMI) from this study to reduce heterogeneity among controls. In ADNI, where longitudinal methylation profiles are available, we kept baseline and second visit samples only (Supplementary Data 1, 2, Methods).

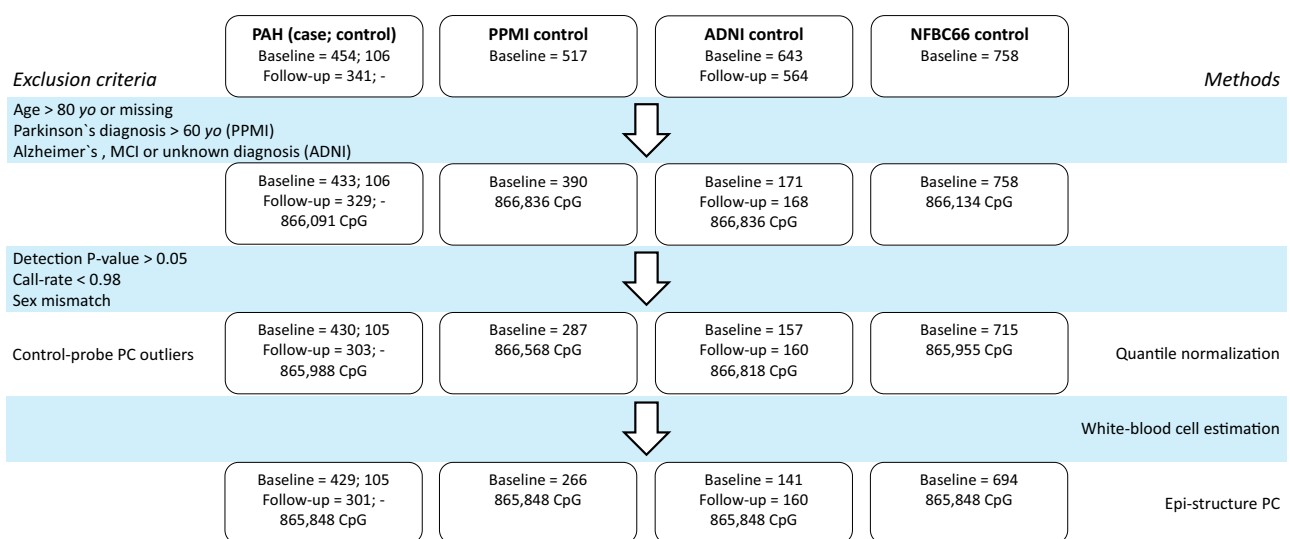

**Fig. 1 | DNA methylation data quality control. Raw DNAme data from the PAH Cohort Study and three external datasets were processed following the published CPACOR pipeline.** ADNI Alzheimer's Disease Neuroimaging Initiative, NFBC1966 Northern Finland Birth Cohort 1966, PPMI Parkinson's Progression Markers Initiative, MCI mild cognitive impairment, PC principal component.

**Table 1 | Demographics and clinical characteristics of individuals with PAH in the EWAS**

| | | UK PAH Cohort Median (25–75%) |
|---|---|---|
| Idiopathic/heritable PAH, n | | 389/40 |
| Age at sample | | 51.27 (40.27–64.03) |
| Sex, n | Female | 314 |
| | Male | 115 |
| WHO functional class, n | I | 8 |
| | II | 76 |
| | III | 273 |
| | IV | 57 |
| 6-min walk distance, m | | 336.00 (230.75–414.25) |
| Mean pulmonary artery pressure, mmHg | | 54.00 (46.00–62.00) |
| Mean pulmonary artery wedge pressure, mmHg | | 10.00 (7.00–12.00) |
| Mean right atrial pressure, mmHg | | 8.70 (6.00–12.00) |
| Cardiac index, L/min/m² | | 2.09 (1.68–2.58) |
| Pulmonary vascular resistance, dyn · s · cm⁻⁵ | | 945.45 (636.77–1289.90) |
| *Treatments* | | *Frequency (%)* |
| No PAH targeted therapy | | 41 (10%) |
| Monotherapy | | 134 (31%) |
| Dual therapy | | 217 (51%) |
| Triple therapy | | 37 (9%) |
| Anticoagulation | | 310 (72%) |
| Calcium Channel Blocker | | 74 (17%) |
| Endothelin Receptor Antagonist | | 247 (58%) |
| Prostacyclin analogue | | 96 (22%) |
| PDE5 inhibitor | | 329 (77%) |
| Selexipag | | 11 (3%) |
| Riociguat | | 16 (4%) |
| *Ethnicity* | | |
| White | | 375 (87%) |
| Black | | 8 (2%) |
| South Asian | | 16 (4%) |
| Other ethnicity/not stated | | 30 (7%) |

*PDE5* phosphodiesterase-5.

## EWAS for PAH

We performed a case-control association analysis of DNAme at 865,859 CpG markers in 429 individuals with PAH (Table 1) and 1206 control individuals without PAH (Fig. 1). Three CpG markers reached epigenome-wide significance ($p < 10^{-7}$ Fig. 2, Supplementary Fig. 2) after adjusting for genomic inflation, with 865 CpGs meeting the unadjusted $p < 10^{-5}$ suggestive threshold (Supplementary Data 3, 4). The most significantly associated CpG marker (cg04917472, $p = 7.3 \times 10^{-11}$,) is located ~2 kilobases upstream of the transcription start site (TSS) of *CTSZ* (cathepsin Z) on chromosome 20 and is hypermethylated in PAH samples (OR[95% CI] = 1.495[1.325–1.687] per % increase in methylation, PAH median raw betas across arrays: 50.7–62.1% Fig. 3). There were also several nominally significant CpGs in close proximity to this probe with consistent direction of effect, confirming hypermethylation in this region (Supplementary Data 4). Cathepsin Z (CTSZ) is a member of the cysteine cathepsin protease family, comprising 11 members in humans, with important roles in regulating the extracellular matrix (ECM) via integrin interactions[18]. ECM alterations are a pathological feature of PAH and have been targeted by approaches to manipulate cathepsin S in PAH (see Discussion). The other two differentially methylated CpG probes, including cg27396197:OR[95% CI] = 1.337[1.224–1.462], PAH betas:43.9–54.6% and cg03144189: OR[95%CI] = 1.439[1.287–1.607], PAH betas:59.8–70.1% (Supplementary Figs. 3, 4) are located on chromosome 13 and 1, respectively downstream of *COG6* (Component Of Oligomeric Golgi Complex 6) and upstream of *ZNF678* (Zinc Finger Protein 678). We performed sensitivity analyses of DNAme association with PAH by sex for the lead associated markers and observed similar differences in methylation in females and males analysed separately (Supplementary Fig. 5). Methylation levels at the three significant CpG markers were similar in samples of PAH individuals and controls taken 6–12 months apart, suggesting a certain stability to this mode of gene regulation as PAH progressed (Supplementary Fig. 6). There were no significant differences between the methylation levels of the top three CpG markers in subtypes of PAH (heritable vs. idiopathic), although all three remain hypermethylated compared to controls in both subtypes analysed (Supplementary Fig. 7).

## Methylation of CpGs near known PAH genes

For 16 established genes underlying heritable PAH and catalogued by Southgate et al.[19], we checked the methylation profile of PAH individuals versus controls to define a CpG marker (cg10976975) within the 5′ untranslated region of *BMP10* showing elevated methylation in PAH (Supplementary Data 5, Supplementary Fig. 8) with a Q-value of 0.0013 (OR [95%CI] = 1.26[1.14–1.38]). DNAme markers near *BMPR2* were not affecting PAH risk (Supplementary Data 5, Supplementary Fig. 9). The lack of significant alterations in DNAme in sites located at the other PAH genes, including *BMPR2*, suggest future investigations could be directed towards cell type-specific DNAme in pulmonary vascular cells and lung or cardiac tissues.

## Downstream molecular consequences of CpG methylation at the significantly associated sites

To determine the genes most affected by our methylation signal we analysed the expression of genes present in topologically associated domains (TADs) of the three epigenome-wide significant PAH-associated CpG markers (Methods). TADs were defined from HiC chromatin interaction data in human microvascular endothelial cells (HMEC) and human umbilical vein-derived endothelial cells (HUVEC) for best association with our tissues of interest – the pulmonary endothelium being the primary site of injury in PAH. Gene expression data were only available in PAH patients. Of the 29 tested associations (7–12 for each of three CpG/TADs), the CpG marker (cg04917472) near the TSS of *CTSZ* showed the most marked (strongest effect) associations with *CTSZ* itself (β/SE = −0.27 +/− 0.0001 per Transcripts Per Million/TPM, $p = 0.0167$, Fig. 4A) and *TUBB1* (β/SE = 0.15 +/− 0.0001 per TPM, $p = 0.0124$) expression levels in PAH (Supplementary Data 6).

## Distribution of CTSZ expression in human tissues

*CTSZ* expression is highest in the aorta (median TPM = 566.8, N = 432), in the lungs (median TPM = 443.6, N = 578) and in the coronary artery (median TPM = 427.9, N = 240) according to bulk gene expression data published by The Genotype-Tissue Expression (GTEx) project (Supplementary Fig. 10) [30]. Single-cell RNAseq data from the GTEx Portal available from the left ventricle of the heart as well as from the left upper lobe of the lungs suggests CTSZ is most highly expressed in macrophages but can also be detected in endothelial cells in both tissues (Supplementary Fig. 11).

## CTSZ CpG is hypermethylated in PAH endothelial cells

To determine whether the altered DNA methylation at *CTSZ* we observe in blood cells is present in the pulmonary endothelium, we queried public DNAme data in pulmonary endothelial cells from PAH samples (Methods). These demonstrated a trend towards hypermethylation at the *CTSZ* CpG cg04917472 ($p$-value = 0.0773), most clear in heritable PAH (HPAH) cases, thus providing a consensus

**a**

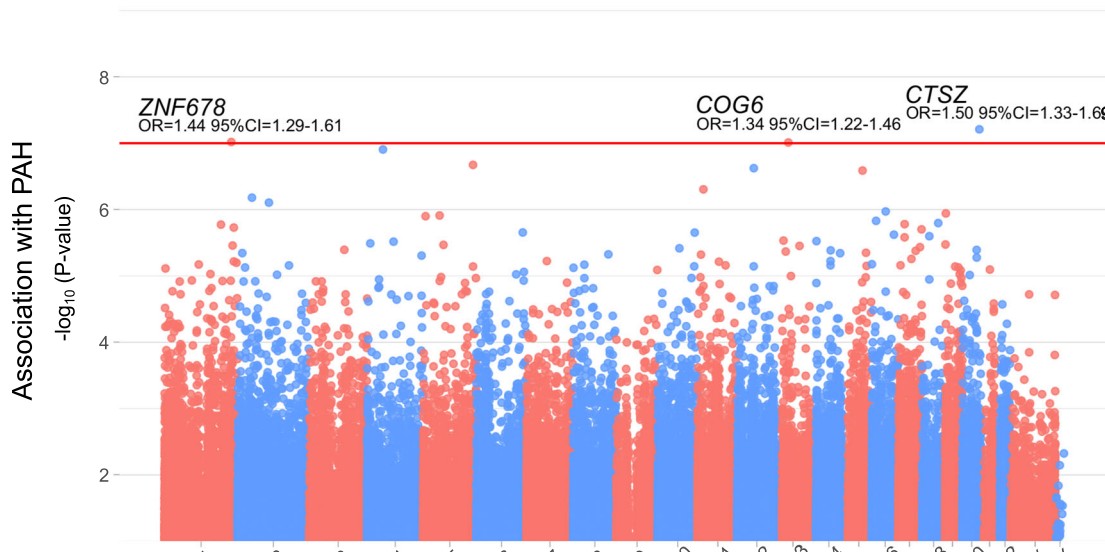

**b**

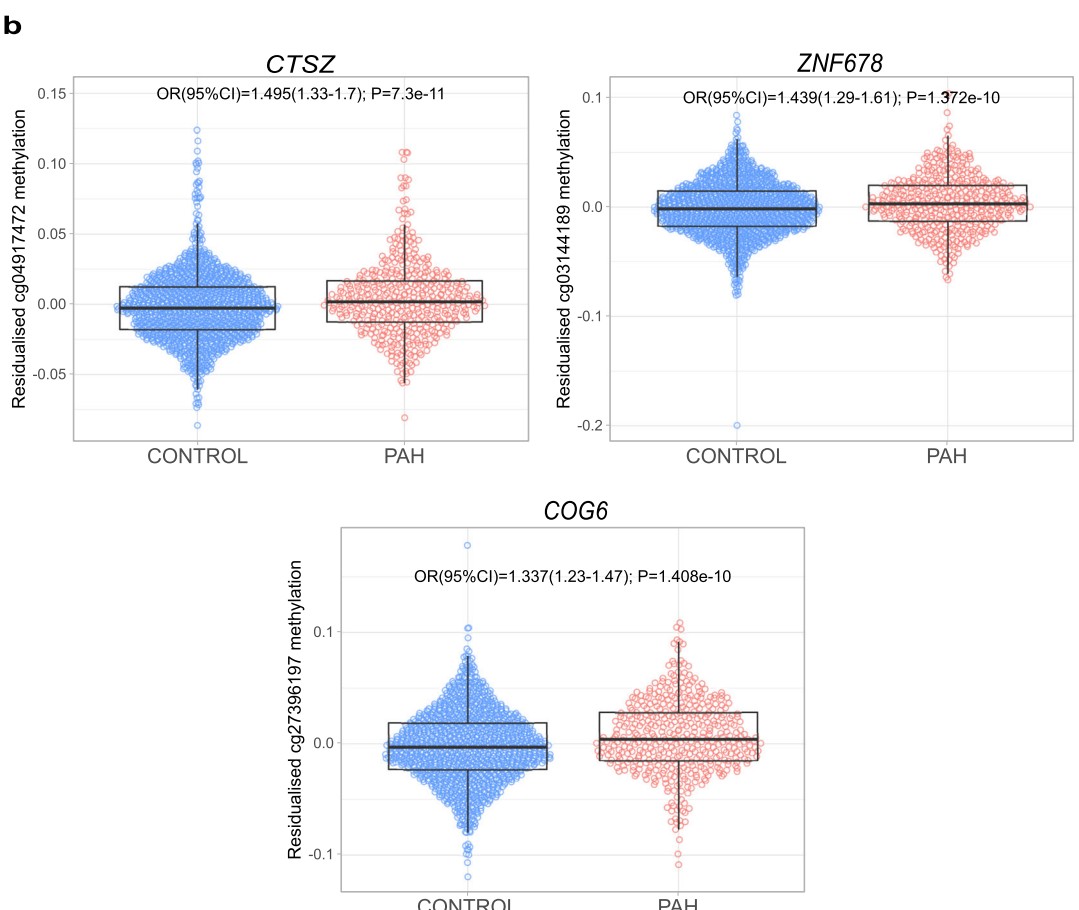

**Fig. 2 | Epigenome-wide association study (EWAS) of PAH. a** Manhattan plot of EWAS of PAH. DNAme markers are ordered according to their genomic positions along the x-axis. Inflation-adjusted *P*-values for the CpG marker effect are plotted along the y-axis on the -log₁₀ scale. The red horizontal line corresponds to the epigenome-wide significance threshold of $P < 10^{-7}$ with annotated markers reaching this threshold. **b** Methylation levels of the three PAH-associated DNAme markers in 429 individuals with PAH and 1226 controls. Residuals of CpG methylation levels after adjusting for covariates in the EWAS are plotted along the y-axis. Boxplots show the median, interquartile range (IQR) and whiskers extend to 1.5 times the IQR. PAH odds ratio (OR) and 95% confidence intervals from multivariable regression analyses are shown, with unadjusted *p*-values in **b**.

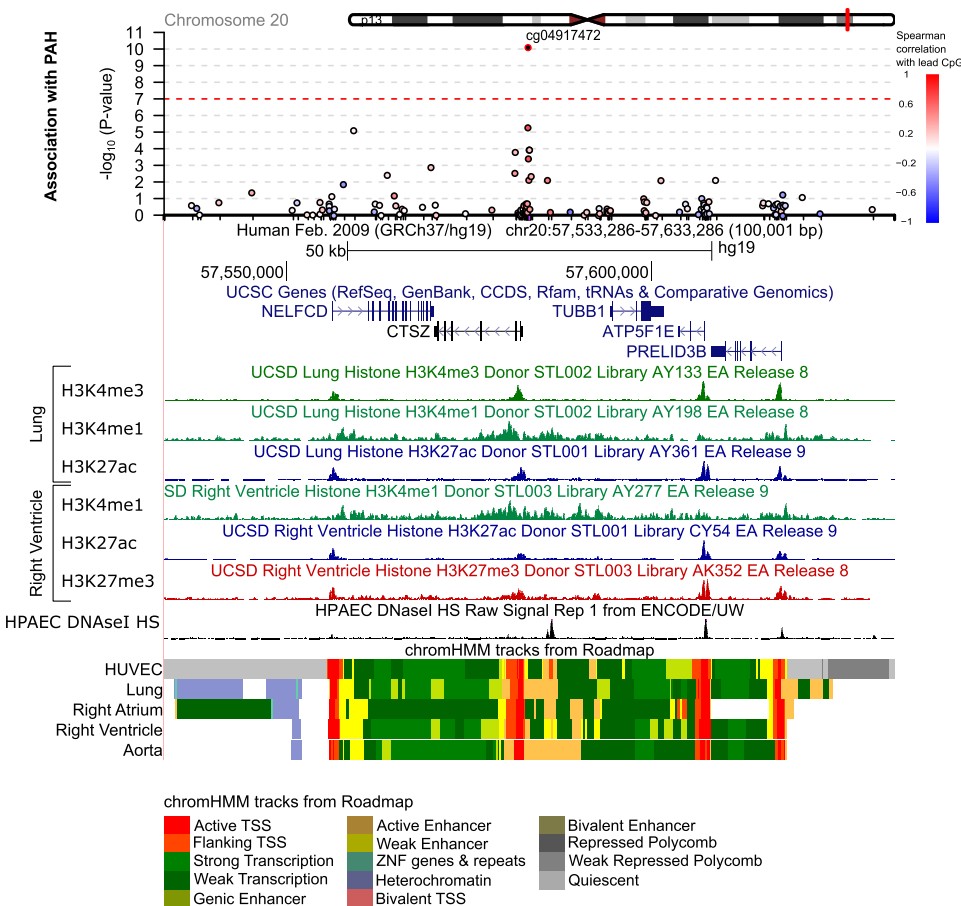

**Fig. 3 | Genomic region containing the significant DNAme marker near *CTSZ* identified by the epigenome-wide association study (EWAS).** A 50-kilobase region on either side of the top DNAme marker (cg04917472) are shown. Each circle represents a DNAme marker coloured by its correlation value with the top DNAme marker (only markers with significant [$P < 0.05$] Spearman's rank correlation coefficients are coloured; red-positive, blue-negative). Epigenomic data in endothelial cells including pulmonary artery endothelial cells (HPAEC) and human umbilical vein-derived endothelial cells (HUVEC), lung and the right heart indicate areas likely to contain active regulatory regions and promoters. Markers include histone H3 lysine 4 monomethylation (H3K4Me1; often found in enhancers) and trimethylation (H3K4Me3; strongly observed in promoters) and H3 lysine 27 acetylation (H3K27Ac; often found in active regulatory regions). Auxiliary hidden Markov models, which summarise epigenomic data to predict the functional status of genomic regions in different tissues or cells, are shown (chromHMM). Red marks signal active transcription start sites. Annotation data were extracted from the UCSC website[62,63]. The UCSC Session URL is available in the Supplementary Materials section.

between DNAme levels in our data and a PAH-relevant cell type ($p = 0.0311$ HPAH vs controls, Supplementary Fig. 12). Hypermethylation was observed for the *BMP10* CpG ($p < 0.05$ for both PAH and HPAH, Supplementary Fig. 13) but no data were available for the other lead CpG.

### Potential upstream factors determining altered DNA methylation profiles in PAH

We investigated the whole blood RNA profiles of epigenetic regulators and for those PAH patients in whom both RNA and DNA methylation data were available assessed the correlation between these regulators and the three lead CpG markers. We found that expression levels of DNA methyltransferase DNMT3A and histone deacetylases HDAC1/2/6/11 were significantly different between PAH and controls (all $p < 0.05$) and that there were significant correlations between the CpG markers and several of the epigenetic regulators (e.g. cg04917472 vs TET2 Rho $= −0.29$, $p = 1.32 \times 10^{-7}$, Supplementary Data 7).

### Protein levels of CTSZ in PAH patient samples

To elucidate whether the altered methylation and gene expression of CTSZ was reflected at the protein level, we queried proteomic data from the UK PAH Cohort Study (Methods) and found the levels of plasma CTSZ to be elevated in PAH compared to healthy controls (OR [95%CI] 1.47 [1.17–1.87], $p = 0.0012$, Fig. 4B). This confirmed that CTSZ was associated with the disease, but the direction of change suggested that white blood cells (which we predict would contain lower CTSZ protein following reduced mRNA) might not be the primary determinant of (elevated) systemically circulating protein. To investigate this in the primary site of disease pathology we performed immunohistochemical staining for CTSZ protein. In lung tissues from controls and end-stage PAH patients (Methods) following lung transplant, CTSZ expression was strongest in airway macrophages and the pulmonary endothelium, particularly in the pathologic hallmark network-like plexiform lesions (Fig. 4C). Some staining was found in the media of PAH patients not observed in controls.

### Associations between circulating CTSZ protein and pathobiological markers

To better understand the potential reasons for elevated circulating CTSZ protein in PAH we performed a linear regression analysis against markers of PAH disease features including renal dysfunction, inflammation and cardiac stress. We defined significant positive correlations between plasma CTSZ levels and renal marker cystatin C, inflammatory cytokine IL-6 and negatively with cardiac marker NT-proBNP (all

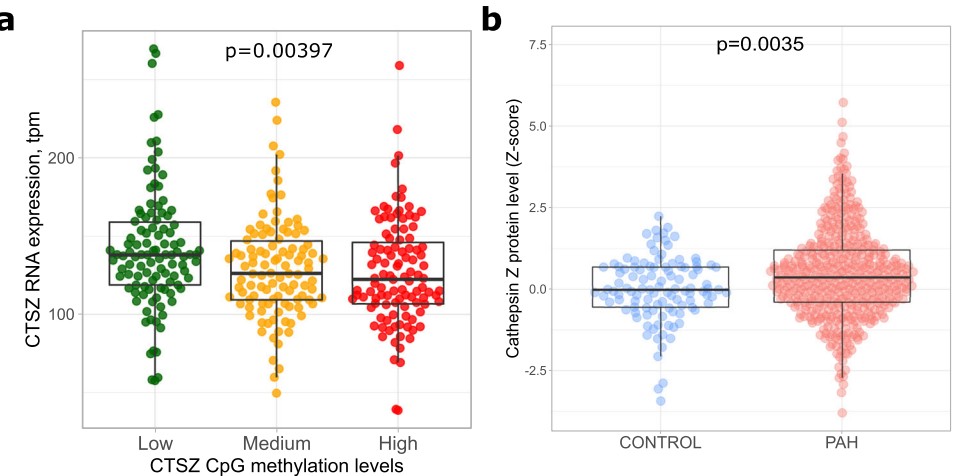

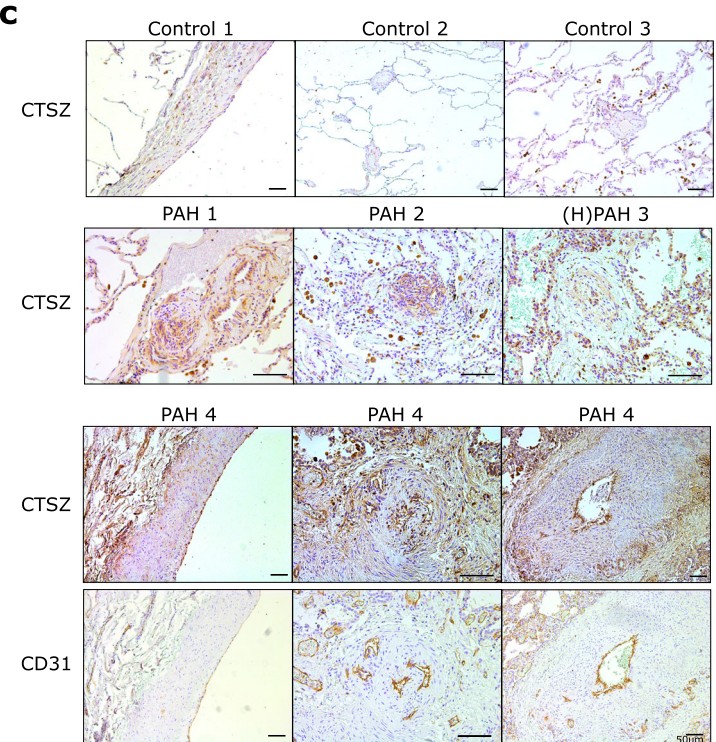

**Fig. 4 | Cathepsin Z in PAH patients. a** Blood RNA CTSZ levels in PAH patients divided by CTSZ CpG methylation status $n = 276$ patients were divided into tertiles of CpG methylation levels for comparison with mRNA levels by Kruskal–Wallis one way ANOVA. **b** Plasma levels of cathepsin Z in PAH versus controls, Logistic regression models used to test the effect of cathepsin Z on PAH were adjusted for age, sex, first two principal components and coagulation factor X. $n = 463$ PAH cases and 108 disease-free controls. Boxplots depict median, first and third quartiles (Q1/Q3 = IQR), and whiskers indicate Q1/Q3 +/− 1.5 * IQR. **c** Immunohistochemistry of CTSZ in PAH lung tissue. CD31 is shown as a marker for endothelium. $N = 3$ vs 4 control and PAH patient tissue sections. Scale bar represents 50um.

$p < 0.05$, Supplementary Data 8). It may be that in the early stages of disease, decreased CTSZ is associated with hypermethylation of the promoter, but later in the disease inflammation and renal dysfunction elevate CTSZ levels. Consistent with this, CTSZ RNA expression in right ventricle tissue (RV) from non-failing controls ($n = 14$), and patients with compensated ($n = 11$) were lower than those in late-stage decompensated ($n = 7$) RV from public dataset GSE198618 ($p = 0.002$ decompensated RV vs controls, Supplementary Fig. 14).

**Blood CpG profiles for biomarker utility**
We explored relationships of the three CpG loci methylation levels with clinical severity markers including walk tests, cardiac biomarker NT-proBNP and cardiac catheter measures, but found no strong correlations (all Rho < 0.13, Supplementary Data 9).

**Functional impact of CTSZ protein on the pulmonary endothelium, the principal site of injury in PAH**
CTSZ promotes apoptosis in neuronal cells[20] but its knockdown stimulated cell death in gastric cancer lines[21] emphasising the importance of establishing cell type-specific roles of cathepsin function. In PAH, the pulmonary endothelium becomes pro-proliferative, anti-apoptotic and disorganised, partly in response to inflammatory stimuli. We assessed apoptosis, proliferation and cell viability in human pulmonary artery endothelial cells (hPAEC) following *CTSZ* knockdown by siRNA (Fig. 5A, Methods). Efficient knockdown of *CTSZ* was associated with increased activation of caspases-3/7, both in vehicle-treated cells and in response to 24 h TNF-alpha or lipopolysaccharide (Fig. 5B), without significant differences in the number of viable cells between siRNAs (Fig. 5C) indicating an enhanced apoptotic response

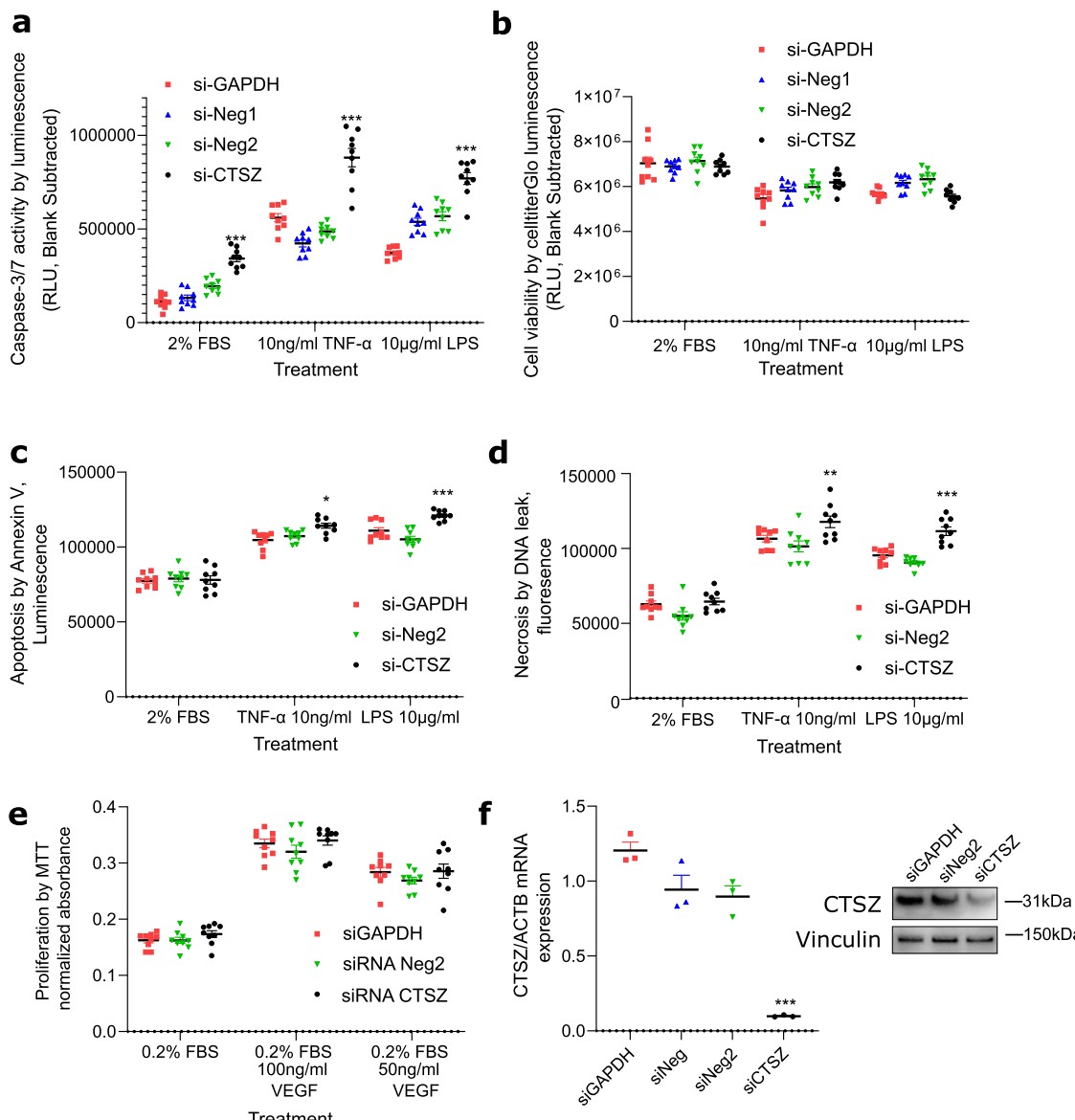

**Fig. 5 | CTSZ effect on hPAEC function by siRNA manipulation. a** Apoptosis activation measured by caspase-3/7 activity in hPAEC following siRNA plus vehicle or indicated pro-inflammatory treatments. **b** Cell viability measured by CellTiterGlo assay in hPAEC 48 h following siRNA treatment and 24 h in 2%FBS-ECM plus vehicle or indicated treatments to match caspase assay. **c** Apoptosis and **d** necrosis measured by Annexin V cell surface binding and DNA leak. *$P < 0.05$, **$P < 0.01$, ***$P < 0.001$ versus both control siRNAs. **e** Proliferation estimated by metabolically active cells in MTT assay following siRNA treatment. $n = 9$ independent cultures from 3 independent experiments for **a**–**e**. **f** CTSZ expression (compared to loading controls vinculin and beta-actin) following siRNA knockdown in hPAEC by immunoblot and qPCR. siNeg negative control scrambled siRNAs, siGAPDH control siRNA targeting GAPDH. $n = 3$ cell cultures from 3 independent experiments. Bars indicate mean and SEM. One way ANOVA with Dunnett's posthoc tests corrected for multiple comparisons in all figures. ***$p < 0.0001$ vs. other siRNAs within treatment group.

specifically associated with loss of *CTSZ*. This was consistent with increased Annexin V translocation to the cell surface (Fig. 5D) and DNA leakage indicating necrosis following apoptosis. Proliferation at baseline and in response to VEGF was unaffected by loss of CTSZ following siRNA treatment as determined by tetrazolium dye (MTT) assay (Fig. 5E). Our in vitro data, presented here, indicate that dysregulation of CTSZ in hPAEC drives hypersensitivity to inflammatory stimuli triggering apoptosis of ECs, a key early event in the pathology of PAH.

## Discussion

Our report describes the first large-scale epigenome-wide estimation of the variability in DNA methylation in whole blood samples in individuals with PAH, where cases were defined by the expert centres across the UK. We report three lead CpGs differentially methylated in PAH, the most significant of which locates in close proximity to the gene encoding Cathepsin Z, *CTSZ*, and two more closest to *COG6* and *ZNF678*. We further demonstrated that the predicted loss of *CTSZ* expression, associated with hypermethylation, alters the function of hPAEC, driving enhanced apoptosis in response to inflammatory stimuli, a characteristic feature of early PAH.

Hypermethylation of the *CTSZ* CpG site (cg04917472) was also observed in PAEC derived from heritable PAH cases in an external study[22]. Previous reports have revealed a correlation between high CTSZ expression levels and advanced malignancy in hepatocellular carcinomas and colorectal, gastric, and prostate cancer[23–26]. The tumour-promoting functions of CTSZ are not dependent on its described catalytic activity but instead are mediated by the Arg−Gly−Asp (RGD) motif in its pro-domain. Impaired proliferation in

CTSZ[-/-] cells was restored by overexpression of CTSZ with a mutated enzymatic site but not mutant RGD, which regulates interactions with integrins (integrinB3 blocking antibodies reduce proliferation and invasion stimulated by CTSZ) and the extracellular matrix[27].

CTSZ also has an inhibitory effect on VEGF-mediated endothelial function in human umbilical vein-derived EC; VEGF-induced EC spheroid sprouting and invadosome formation were not significantly altered by targeted cathepsin B inhibition alone, but significantly suppressed when both CTSZ and B were inhibited[28]. The confirmation of altered DNAme at the CTSZ-associated CpG we identified in heritable PAH patient-derived PAEC through our analysis of public methylation data demonstrates that our approach has the potential to identify both blood cell-specific and systemic changes using a non-invasive sampling method. Blood analysis allows for sampling a larger cohort to derive meaningful associations, enabling relevant targeted investigations in pulmonary vascular cells or lung tissue.

Recent investigation into the role of cathepsins in idiopathic PAH has focused on increased expression of cathepsin S – a potent elastase - in smooth muscle cells; this was associated with decreased elastin content and degradation of the elastic laminae of pulmonary arteries[29] and stiffening of the arterial walls. In the same study, administration of a selective cathepsin S inhibitor (M249314) to rats with monocrotaline-induced PH prevented the degradation of elastic laminae and hypertrophy of PAs, suggesting that cathepsin S plays a role in the pathogenesis of PH. M249314 reduced cathepsin S levels by activating PPARγ, a nuclear receptor with antiproliferative and proapoptotic functions shown to cause severe PAH[30]. CTSZ is a shared PPARα/γ target gene and a regulator of brown adipocyte thermogenic function, but the effect of CTSS inhibition on CTSZ has not been measured[31]. Treatment with elastase inhibitors has proven effective in reversing advanced remodelling of the PAs and normalising PA pressure in MCT-PH rats[32]. A phase I clinical study investigating the safety of subcutaneous Elafin, an elastase inhibitor that inhibits neutrophil elastase, as a treatment for severe PAH has been completed (ClinicalTrials.gov:NCT03522935), highlighting the translational potential for targeting elastase and cathepsin pathways in PAH[33,34].

COG6, which is closely located to the second-most significant DNAme site (cg27396197), has not been previously linked with PAH; biallelic mutations in COG6 are associated with congenital disorders of glycosylation, which are related to defects in the synthesis of glycans and their attachment to glycan proteins and lipids[35]. There are marked differences in the levels of sugars, amino sugars, and nucleotide sugars between PAH PASMCs and control cells, suggesting altered protein and lipid glycosylation capacity and polysaccharide biosynthesis is relevant to the PAH disease state[36]. PAH PASMCs also demonstrated a reduction in the levels of fructose and glucose-6-phosphate, metabolites that serve as universal intermediates in carbohydrate pathways related to glycosylation. ZNF678 is the closest gene to the third CpG identified in this study (cg03144189). While not previously related to PAH, other zinc finger containing transcription factors, including ZNF430, ZNF638, ZNF577 and ZNF28, were recently found to be differentially expressed in PAH whole blood samples[37].

Of the 16 genes so far linked to heritable, idiopathic and childhood-onset PAH[19], we were able to identify DNAme changes in one, BMP10, 5' untranslated region. The mutations in BMP10 in idiopathic PAH are truncating and predict loss-of-function[38]. BMP10 is a close paralogue of BMP9 that encodes an activating ligand for ALK1 (ACVRL1) and can substitute for BMP9-associated angiogenic signalling in BMP9 knockout mice[39,40]. BMP9 deficiency is associated with the development of PAH and BMP9 is under development as a therapeutic strategy[41]. BMP9:BMP10 heterodimers have been shown to circulate in human plasma and represent the major biologically receptor-activating complex[42]; changes in BMP10 expression levels could impact BMP9 signalling through this activator complex.

Clustering of DNAme profiles from cultured pulmonary endothelial cells identified two clusters of DNAme sites that discriminated PAH from controls[22]. Hypomethylation of GNLY, which encodes granulysin, in human peripheral blood mononuclear cells and explanted lungs distinguished PVOD from PAH[43]. DNA methylation of the proximal promoter of endothelial NO synthase (eNOS) negatively impacts functional activity as well as vascular tone and angiogenesis, showing regulatory roles for DNA methylation in a gene constitutively expressed in the vascular endothelium[44]. Previous comparative analysis of whole epigenome-wide DNA methylation profile of cultured PASMCs from CTEPH and control cell lines detected 6829 significantly differentially methylated probes (DMPs) between these two groups. Among these, 4246 DMPs were hypermethylated, while 2583 DMPs were hypomethylated[9]. Our observations are consistent in that we found in this study that more CpGs are hypermethylated in PAH. We were also able to demonstrate that the changes in DNAme at the three loci associated with PAH were stable across samples taken one year later than the baseline analysis, providing evidence that these alterations are long term and could continue to influence disease progression.

Although much enthusiasm surrounds the field of epigenetics and its promise in understanding disease aetiology, best practice and methods for EWA study design and analysis are in their infancy compared to genome-wide association studies (GWAS). Current EWA study design and methods are largely similar to those of GWAS, despite important differences between germline genetic- and epigenetic variation, such as tissue specificity and malleability. For example, selection biases and confounding should be considered when designing EWAS, given the dynamic nature of the epigenome. Correcting for obvious confounding factors – such as age, sex, smoking and cell fractions – can alleviate this problem, but residual confounding due to unknown factors likely remains. EWAS studies do not routinely report or adjust for test statistic inflation which can be as high as $\lambda = 20$ for some publicly available EWAS results[45]. To reduce false positive findings in our study, we corrected test statistics for the inflation parameter ($\lambda = 1.45$) and applied a significance threshold suggested to adequately control for the false positive rate of Illumina EPIC array studies[46]. Given the tissue-specificity of epigenetic variants, disease-associated regions or markers identified in whole blood should be subsequently tested in (other) disease-relevant tissues, which we implemented by using both public and our own in vitro data from the pulmonary endothelium. There is a SNP (rs114059951) located within one base pair of the cg04917472 CpG; in The Genome Aggregation Database (gnomAD) rs114059951 is reported with a minor allele frequency - MAF of 0.003535 and therefore is going to be present in a very few individuals and would not materially change our findings. Lastly, it is important to note that while the EPIC array was the highest-coverage DNAme array platform available at the time of analysis, it covers <4% of ~28 million CpGs and could miss important disease-associated markers measurable with the considerably more expensive whole-genome bisulphite sequencing. BMI and diabetes information were not available for all studies and thus are not corrected for. A priority for future studies would be validation of our findings in a distinct patient cohort, for example by targeted DNA methylation/pyrosequencing analysis. Future work will also pursue the analysis of DNA methylation on a regional or per-gene basis rather than at the single CpG level. The role of CTSZ in interacting with integrins expressed on platelets and activated endothelium and VEGF led us to focus on its effects on PAEC function but future efforts should establish its potential role in PASMC. Our analysis of circulating CTSZ levels demonstrated elevated levels in contrast to the associated reduction in RNA driven by hypermethylation, but further analysis showed circulating levels reflected a combination of renal function, inflammation and cardiac stress. Circulating protein levels may reflect release from dying cells and dysregulation in multiple tissues beyond the blood and endothelium. Interestingly, CTSZ RNA levels were elevated in RV tissue of decompensated patients

supporting a distinction between early and late-stage disease in CTSZ regulation.

We have demonstrated that PAH is associated with altered blood DNA methylation profiles, exemplified by the protease cathepsin *CTSZ*, which modifies pulmonary endothelial function. Many other signals are reported which require further analysis in collaborative studies. Cathepsins are druggable and dysregulation in circulating cathepsin Z presents a priority pathway for therapeutic investigation in PAH.

## Methods

### UK PAH Cohort study

Patients with idiopathic or heritable pulmonary arterial hypertension (ihPAH) and healthy controls were recruited to the UK PAH Cohort study after providing written, informed consent. The study was approved by local ethics committees (East of England Research Ethics Committee [REC] 13/EE/0203).

### External controls

**Parkinson's Progression Markers Initiative (PPMI).** PPMI is an ongoing observational, international, multicentre (16 US and 5 European sites) study launched in 2010 aimed to identify serological, genetic, spinal fluid and imaging biomarker of Parkinson's disease progression in a large cohort of newly diagnosed individuals compared to matched healthy controls[47]. Genomic DNA was isolated from 200–400 µl whole blood samples using the QIAamp DNA Blood Mini Kit according to the manufacturer's instructions. Purity of DNA was assessed based on Abs 260/ Abs 280 > 1.8 and Abs 260/ Abs 230 > 2.0 ratios using Nanodrop. The extracted DNA was quantified using Quant-iT PicoGreen (Thermo Fisher cat. no. P7589), or with Qubit HS dsDNA assay (Thermo Fisher cat. # Q32854), with samples run in duplicate following the manufacturer's protocol. All samples were blind coded and randomised for array processing. Genomic DNA (500 ng$^{-1}$ ug) was bisulfite converted (EZ DNA Methylation kits, Zymo Research, D5003) per Illumina's recommendation. PCR cycle conditions were adjusted according to Illumina's specification, as indicated in both Zymo's and Illumina's protocols. The samples were processed and hybridised to the Illumina Infinium Human MethylationEPIC BeadChip Array. BeadChips staining steps were automated with Illumina's Tecan robot. The BeadChips were scanned with Illumina's iScan. DNA methylation profiling is registered under PPMI Project ID 140.

**Alzheimer's Disease Neuroimaging Initiative (ADNI).** Data used in the preparation of this article were obtained from the ADNI database (adni.loni.usc.edu). The ADNI was launched in 2003 as a public-private partnership with the primary goal of testing whether serial magnetic resonance imaging, positron emission tomography, other biological markers, and clinical and neuropsychological assessment can be combined to measure the progression of mild cognitive impairment and early Alzheimer's disease. Clinical descriptions of the cohort and detailed methods for profiling DNA methylation have been published[48,49]. In brief, longitudinal whole-genome DNA methylation profiling in peripheral blood samples of 653 ADNI participants selected from two phases of ADNI (ADNI2 and ADNIGO) was performed using the Illumina Infinium Human MethylationEPIC BeadChip Array. Samples were randomised using a modified incomplete balanced block design, whereby all samples from a subject were placed on the same chip, with remaining chip space occupied by age- and sex-matched samples.

**Northern Finland Birth Cohort 1966 (NFBC1966).** NFBC1966 comprises participants from the two northernmost provinces of Finland with expected dates of birth falling in 1966 (*n* = 12,058 births), data deposited at: https://etsin.fairdata.fi/dataset/716939c3-7a2a-4b6a-91f3-92aca09bc52d. Cohort profile and data collection details have been published[50]. From the medical examination timepoint at 46 years,

DNA methylation data were measured on the Illumina Infinium MethylationEPIC BeadChip Kit array for 758 selected subjects that attended the clinical examination, completed the questionnaire and had DNA methylation data from the previous clinical examination at 31 years available. Consent was obtained and the study was approved by the ethical committees of the University of Oulu and Imperial College London (Approval:18IC4421).

### Methylation assays, data QC and normalisation

DNA methylation profiles of peripheral blood were assayed using the Illumina Infinium MethylationEPIC BeadChip Kit which covers over 850,000 methylation sites. The *CPACOR* pipeline published by Lehne et al.[17] was used for the processing on genome build GRCh37/hg19, quality control (QC) and normalisation of the raw methylation data (Fig. 1). Staining and hybridisation checks were performed using Illumina's GenomeStudio software v1.0 and were omitted from the *CPACOR* QC. For each batch (in case of the PAH Cohort Study) and dataset (in the case of external cohorts ADNI, PPMI and NFBC1966) separately, we removed CpG markers with detection *p*-value > 0.049 in over 50% of samples by setting their values to missing for all samples. For each plate and dataset separately, we removed samples with <98% of CpG markers successfully called (detection *p*-value > 0.049). Samples discordant for reported and genetic sex, based on CpGs on the X- and Y-chromosome, were also excluded from the study. We then quantile-normalised intensity values for all batches and datasets concatenated together followed by the estimation of white blood cell type subpopulations based on 100 CpG sites by the Houseman method[51] as implemented in the *minfi* R package v.1.30.0[52]. Additionally, we excluded outlying samples based on the top four principal components (PC) of the autosomal, quantile-normalised DNA methylation data, by three times the standard deviation per PC. Ancestry-related principal components were calculated with the EPISTRUCTURE method[53] implemented in GLINT software v.1.0.4[54]. Smoking status was predicted using the EpiSmokEr: Epigenetic Smoking status Estimator method v.0.1.0[55]. CpG marker annotations were obtained using the *IlluminaHumanMethylationEPICanno* R package v.0.6.0[56].

### Epigenome-wide association study

We evaluated the association between DNA methylation level and diagnosis in multivariable regression models in up to 1635 individuals including 429 PAH individuals at each of 865,848 CpG markers. The UK PAH Cohort Study has a higher proportion of females (self-reported sex) compared to external control cohorts, in keeping with the female predominance in this disease (Supplementary Data 1). There was good concordance between smoking status predicted from the DNA methylation data and smoking status reported in subsets of the PAH and NFBC1966 cohorts (Supplementary Fig. 1). There is significant variance between the UK PAH cohort and the external ADNI and PPMI control groups in terms of proportions of predicted smoking status and four out of six white blood cell fractions (Supplementary Data 1), therefore these were adjusted for in the EWAS model. Quantile-normalised beta values at each CpG marker were tested for their association with PAH whilst adjusting for sex, age, estimated white blood cell fractions and predicted smoking status, first ten principal components computed from control probes and the first five "epi-structure" principal components to adjust for batch and ancestry differences. In this study, CpG markers reaching an epigenome-wide threshold ($p < 10^{-7}$) after adjustment for the genomic inflation factor ($\lambda = 1.45$) were considered statistically significant. Markers reaching a nominal $p < 10^{-5}$ are reported in the supplement as suggestively associated for future study assessments. One-year follow-up samples were available for a subset of 301 PAH samples and 160 ADNI controls which we used to assess the stability of significant CpG markers over time (Supplementary Data 2). 557 CpG markers at 16 established genes underlying heritable PAH catalogued by Southgate et al.[19] were

assessed in a targeted sub-analysis reported with a FDR-corrected q threshold based on the number of markers tested.

For external confirmation of the relevance of our findings to pulmonary vascular cells, public data from Hautefort et al.[22] in pulmonary endothelial cells from individuals with PAH were downloaded and analysed for our lead CpG marker, compared to control data from the same study, as well as sub-analysis by heritable and idiopathic PAH subtype.

## Transcriptomics and proteomics

Plasma proteomics (SomaLogic SomaScan v4) and whole blood transcriptomic (RNAseq) data were available from the UK PAH Cohort study[37,57].

## Transcriptomics

Whole blood (3 ml) was collected in Tempus™ Blood RNA Tubes, which were stored at −80 °C until required. RNA was extracted using a Maxwell robotic system (Promega). Samples with a 260/230 ratio >1.5 and a 260/280 ratio in the range 1.9–2.1 were further quality checked by Bioanalyser and those achieving a minimum RNA Integrity Number of 7 were submitted for sequencing. Globin-Zero Gold rRNA Removal Kits (Illumina Inc, San Diego, CA) were used to remove ribosomal RNA contamination from whole blood RNA samples. 75 bp paired-end sequencing on a Hiseq4000 was performed on pooled libraries of ~80 samples. Fastq files (raw reads from RNAseq) were analysed using Salmon v0.9.1[58] and GENCODE release 28 to produce transcript abundance estimates which were converted to gene expression data using tximport in R[59]. Estimated gene abundances (in TPM) were then analysed in comparison to residuals of DNAme levels – betas adjusted for the EWAS covariates – for the lead CpGs against nearby genes within topologically associated domains (see below) correcting for multiple comparisons by FDR.

## Proteomics

Peripheral venous plasma EDTA samples were collected as previously described[60]. Participants were not fasting and were sampled at their routine clinical appointment visits. The plasma samples underwent one freeze-thaw cycle to aliquot 120 μL for SomaLogic SomaScan v4 assay. 4349/5284 somamers targeting 4152 unique proteins were included for analysis following removal of 305 non-human/non-protein aptamers and quality control to select only those with stable measurements defined as <20% coefficient of variance in the repeated pooled plasma assay controls.

Relative fluorescence units were $\log_{10}$ transformed to normalise protein levels, then corrected for the first two principal components (derived from all proteins from the panel) by linear regression to correct for population stratification or sample quality differences. Finally, protein levels were standardised to the control levels (converted to z-scores using the mean and SD in controls as reference) for ease of interpretation of results and comparability of proteins. PAH status was modelled as a function of the protein level, coagulation factor X, age and sex in logistic regression. 463 PAH cases and 108 disease-free controls were available for these analyses after quality control checks and corrections. Multiple comparisons were corrected using Benjamini–Hochberg false discovery rate (FDR).

## Topologically Associating Domains

Topologically Associating Domains (TAD) are determined using the online Topologically Associating Domain Knowledge Base[61]. For each TAD, the 3D structure is predicted using multidimensional scaling method, using protein-coding gene data from Ensembl and three lncRNA databases, including NONCODE 2016, LNCipedia 4.0, and lncRNAdb 2.0. Genes are then mapped onto TADs of the available cell types by comparing their genomic positions and the domain definitions are called using normalised Hi-C data at the resolutions of 50 kb.

For our study, we chose to look at TADs in human microvascular endothelial cells (HMEC) and human umbilical vein-derived endothelial cells (HUVEC) for best association with our tissues of interest.

## In vitro analyses

Human pulmonary artery endothelial cells are obtained from Promo-Cell GmbH (#C-12241 Lot:#458Z016.14) and cultured according to supplier protocols in 10% FBS Endothelial Cell Growth Medium 2 (EGM2; PromoCell #C-22111) at 37 °C, 21% O2, and 5% CO2 in a humidified incubator with medium changes every 48 h. Cells were passaged once they reached 80 to 90% confluence. hPAECs used for experiments were between passages 5 and 8. Silencer Select siRNA ordered from Thermo Fisher Scientific (Negative control no.1: #4390843; Negative control no.2: #4390844; GAPDH positive control: # 4390849; CTSZ: # 4390824s292). Cells are transfected at 80% confluency in 96 well plates according to the Life Technologies Lipofectamine® RNAi-MAX (#13778-075) Protocol for 4 h, before being incubated overnight in 0.2% FBS EGM2. Treatment was subsequently carried out with TNF-α (R&D systems #10291-TA-050) at 10 ng/ml or 10 μg/ml LPS (Sigma #L4516-1MG), all made up in 2% FBS cell media. After an additional 24-hour incubation at 37 °C, 5% CO2, 100 μl of Caspase-Glo 3/7 reagent (Promega) was added per well. After 30 min of incubation, the luminescence signal was determined in a Promega GloMax plate reader. To investigate cell number effects, where caspase activity was low owing to high cell death, a companion CellTiter-Glo assay (Promega) was carried out to measure ATP production as a surrogate of cell viability.

To show that increased caspase activity was associated with cell apoptosis, a parallel validation experiment was carried out post-siRNA and TNF-α and LPS treatment. 100 μl of 2× detection reagent from the RealTime-Glo™ Annexin V Apoptosis and Necrosis Assay (Promega) was added per well. Annexin V binds to externalised phosphatidylserine on apoptotic cells whilst a cell-impermeant, profluorescent DNA dye binds to DNA that had leaked from necrotic cells. Luminescence and fluorescence were measured by Promega GloMax.

To investigate the effects on cell proliferation in hPAEC following siRNA exposure, MTT assays were performed after VEGF [50 and 100 ng/ml] or vehicle was added to hPAEC cultures for 24 h. On the day of the MTT, the EGM2 was replaced, and blank wells were created. 10 μl of MTT [5 mg/ml] reagent was added, and the cells were incubated for 4 h. Following this, 100 μl of MTT detergent (94.65% isopropanol, 5% NP40, 0.35% HCL [1 M]) was added to each well and the plate was shaken for 10 min. A plate reader was then used to test absorbance at 570 nM.

## Immunohistochemistry

Lung tissue sections (deparaffinised) were stained with anti-CTSZ antibody (Abcam, UK ab180580) or anti-CD31 (DAKO-MO823/Clone-JC70A) at 1:200 (in DAKO-S2022) for 12 h after antigen retrieval by boiling in Citrate buffer (pH-6) for 20 mins and blocking with DAKO peroxidase blocking solution (RT 5 min) and horse serum (1:10 in TBS). Novocastra post primary (RE7111) for 45 min was followed by washing, Novolink polymer (7112) and DAB development for 5 min and counterstain with haematoxylin. Samples used for immunohistochemistry are detailed in Supplementary Data 10.

## Regional plot

UCSC Session URL: https://genome-euro.ucsc.edu/cgi-bin/hgTracks?db=hg19&lastVirtModeType=default&lastVirtModeExtraState=&virtModeType=default&virtMode=0&nonVirtPosition=&position=chr20%3A57534353%2D57634353&hgsid=276677120_z5AXaaoUEFOV1EzrBOKCiT1veLCk.

## Reporting summary

Further information on research design is available in the Nature Portfolio Reporting Summary linked to this article.

## Data availability

Data from the UK PAH Cohort study are available upon reasonable request to the access committee cohortcoordination@medschl. cam.ac.uk. The use of the Materials or Data must be for research projects and work that falls under the remit of the National Cohort Study of Idiopathic and Heritable PAH or are collaborative projects with one or more of the Partners – see https://www.ipahcohort.com/ for details. The EWAS summary statistics will be deposited at ewascatalog.org accessible at https://doi.org/10.5281/zenodo.10276821. Data used in the preparation of this article were obtained from the Parkinson's Progression Markers Initiative (PPMI) database (www.ppmi-info. org/access-dataspecimens/download-data). For up-to-date information on the study, visit ppmi-info.org. Data used in preparation of this article were obtained from the Alzheimer's Disease Neuroimaging Initiative (ADNI) database (adni.loni.usc.edu). The Northern Finland Birth Cohort data used in this study are available upon collaboration and formal data request only, please see http://www.oulu.fi/nfbc. The GTEx data used for the analyses described in this manuscript were obtained from the GTEx Portal on 05/11/2021. Source data are provided with this paper.

## Code availability

The code for the CPACOR analysis pipeline was adapted from Lehne et. al.[17] which was developed and written by Benjamin Lehne (Imperial College London) and Alexander Drong (Oxford University). The code for the low-level quality control was developed and written by Alexander Teumer (University Medicine Greifswald/ Erasmus MC Rotterdam). The code was combined into the current pipeline by Pascal Schlosser and Franziska Grundner-Culemann and it is available at https://github.com/genepi-freiburg/Infinium-preprocessing. The method was then extended to EPIC arrays.

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

## Acknowledgements

The BHF/MRC UK National Cohort of Idiopathic and Heritable PAH made this study possible. We thank National Institute for Health Research (NIHR) BioResource volunteers for their participation, and gratefully acknowledge NIHR BioResource centres, NHS Trusts and staff for their contribution. We thank the NIHR Imperial Clinical Research Facility, NIHR Sheffield Biomedical Research Centre, Cambridge NIHR Cardiorespiratory BRC and NHS Blood and Transplant. The views expressed are those of the author(s) and not necessarily those of the NHS, the NIHR or the Department of Health and Social Care. This work was supported by the NIHR BioResource which supports the UK National Cohort of Idiopathic and Heritable PAH; the British Heart Foundation (BHF SP/12/12/29836) and the UK Medical Research Council (MR/K020919/1). This work was supported in part by the British Heart Foundation Centre for Research Excellence award RE/18/4/34215. C.J.R. is supported by BHF Basic Science Research fellowships (FS/15/59/31839 & FS/SBSRF/21/31025) and Academy of Medical Sciences Springboard fellowship (SBF004\1095). NWM is a BHF Professor and NIHR Senior Investigator. A.L. is supported by a BHF Senior Basic Science Research fellowship (FS/18/52/33808). This work was in part funded by the Diabetes UK (BDA number: 20/0006307), the European Union's Horizon 2020 research and innovation programme (LONGITOOLS, H2020-SC1-2019-874739). I.C. is supported by a Royal Society and Wellcome Trust Sir Henry Dale Fellowship: 224662/Z/21/Z. PPMI – a public-private partnership – is funded by the Michael J. Fox Foundation for Parkinson's Research and funding partners, [see www.ppmi-info.org/about-ppmi/who-we-are/study-sponsors]. The investigators within the ADNI contributed to the design and implementation of ADNI and/or provided data but did not participate in analysis or writing of this report. A complete listing of ADNI

investigators can be found at: http://adni.loni.usc.edu/wp-content/uploads/how_to_apply/ADNI_Acknowledgement_List.pdf. We thank all NFBC cohort members and researchers who participated in the 46 yrs study. We also wish to acknowledge the work of the NFBC project centre. NFBC1966 received financial support from University of Oulu Grant no. 24000692, Oulu University Hospital Grant no. 24301140, ERDF European Regional Development Fund Grant no. 539/2010 A31592. The Genotype-Tissue Expression (GTEx) Project was supported by The Common Fund of the Office of the Director of the National Institutes of Health, and by NCI, NHGRI, NHLBI, NIDA, NIMH, and NINDS.

## Author contributions

Conceptualisation – C.J.R., I.P., M.R.W., N.W.M.; Data curation – C.J.R., J.W., E.M.S., A.U., Y.W., H.D., Z.B., M.R.J., J.A., K.H.H.; Formal Analysis & Writing – original draft – C.J.R., A.U., Y.W., I.P.; Data Acquisition, Data Interpretation and Writing – review & editing – A.U., Y.W., H.D., J.W., E.M.S., I.C., E.V., Z.B., M-R.J., J.A., K-H.H., J.G.C., J.L., C.C., L.S.H., J.P.-Z., M.T., S.J.W., D.G.K., R.C., A.L., S.G., N.W.M., M.R.W., I.P., C.J.R. C.J.R., I.P., A.U., Y.W. and H.D. have had access to, and verified, the data used in this article.

## Competing interests

The authors declare no competing interests.

## Additional information

[1]Department of Clinical and Experimental Medicine, University of Surrey, Surrey, UK. [2]National Heart and Lung Institute, Imperial College London, London, UK. [3]Section of Genetics & Genomics, Department of Metabolism, Digestion and Reproduction, Imperial College London, London, UK. [4]VPD Heart & Lung Research Institute, University of Cambridge, Cambridge, UK. [5]People-Centred Artificial Intelligence Institute, University of Surrey, Guildford, UK. [6]MRC Centre for Environment and Health, Department of Epidemiology and Biostatistics, School of Public Health, Imperial College London, London, UK. [7]Center for Life Course Health Research, Faculty of Medicine, University of Oulu, Oulu, Finland. [8]Unit of Primary Care, Oulu University Hospital, Oulu, Finland. [9]Department of Life Sciences, College of Health and Life Sciences, Brunel University London, London, UK. [10]Institute of Biomedicine, Medical Research Center Oulu, Oulu University and Oulu University Hospital, Oulu, Finland. [11]Department of Pediatric Gastroenterology and Metabolic Diseases, Poznan University of Medical Sciences, Poznan, Poland. [12]University College London, London, UK. [13]University of Newcastle, Newcastle, UK. [14]Golden Jubilee National Hospital and University of Glasgow, Glasgow, UK. [15]Royal Papworth Hospital, Cambridge, UK. [16]National PH Service, Royal Brompton Hospital, London, UK. [17]Department of Infection, Immunity & Cardiovascular Disease, University of Sheffield, Sheffield, UK. [18]Sheffield Pulmonary Vascular Disease Unit, Royal Hallamshire Hospital, Sheffield, UK. [19]NIHR Biomedical Research Centre Sheffield, Sheffield, UK. [20]NIHR BioResource for Translational Research, Cambridge Biomedical Campus, Cambridge, UK. [21]These authors contributed equally: Anna Ulrich, Yukyee Wu. [22]These authors jointly supervised this work: Inga Prokopenko, Christopher J Rhodes. ✉e-mail: crhodes@ic.ac.uk

