## [Peer Review File · Nature Communications]

Blood DNA Methylation Profiling Identifies Cathepsin Z
Dysregulation in Pulmonary Arterial HypertensionREVIEWER COMMENTS

Reviewer #1 (Remarks to the Author):

In the manuscript "Blood DNA Methylation Profiling Identifies Cathepsin Z Dysregulation in Pulmonary Arterial Hypertension" by Ulrich and colleagues, the authors tested whether blood-based methylation changes were associated with the rare pulmonary arterial hypertension (PAH). In order to test this, the authors performed an EWAS using the 850K methylationEPIC arrays in a cohort of 429 individuals with PAH and > 1,206 controls, and found that the cg04917472 probe, located upstream of the CTSZ gene was the most significant, and correlated with a decreased expression of the gene. Further functional validation revealed that the knockdown of the gene induced apoptosis, a hallmark of PAH.

General comments: This work is indeed novel, interesting, well-written and includes functional data that brings biological insight from the EWA study. There are some comments that need to be addressed, detailed below:

1. For the statistical analysis, the authors have tested > 865,000 CpG probes, however, there are probes that should be removed prior to the EWAS analysis, such as: cross-hybridizing probes, non-CpG probes, and probes near SNPs, that could confound CpG detection
 - a. For instance, based on supplementary Table 4, there is a SNP (rs114059951) located within one base pair of the cg04917472 CpG. Similarly for the other two CpG sites.
2. The general accepted threshold for the detection p-values is ≥ 0.01 , whereas the authors used 0.049– is there a reason for this?
3. What is the effect? Are these beta values? Would be good to indicate the % methylation in the text for the top CpG sites - for biological insight.
4. It would be good to highlight the results from Supplementary Table 4: that there are several nominally significant CpGs in close proximity to the CTSZ probe, which are consistent in the direction of effect (i.e., hypermethylated)
5. In Figure 4A, the patients were divided into tertiles of CpG methylation (i.e., low, medium and high methylation) – it is not very clear why this was done? What (and why) is the a difference in analysis performed in Figure 4A vs. Supplementary Table 6?
6. In general, more detail should be included regarding the RNA expression vs. methylation in the methods section (in addition to the details in the figure legends)
7. There is a difference in direction of effect between the RNA and protein, which the authors point out. However, this discordance was not fully explained or addressed. For instance, the authors mention: "It may be that in the early stages of disease, decreased CTSZ is associated with hypermethylation of the promoter, but latterly inflammation and renal dysfunction elevate CTSZ levels.", but there is no evidence to support this.
8. The other two probes that were significant also seem to be interesting – for instance the targeted gene TUBB1 by the cg04917472 probe, in addition to the cg27396197 probe targeting COG6 and MRPS31 – was there a reason why these were not also further functionally studied? Or checked whether associated with protein levels?

Minor comments:

1. The authors mention in the discussion: "it is important to note that while the EPIC array is the highest-coverage DNAm array platform available" – if referring to the 850K EPIC array, there is higher (Infinium MethylationEPIC v2.0 BeadChip). Therefore, could simply indicate that the 850K EPIC array only covers only < 4%.....
2. In the sentence: "its effects on PAEC function but future efforts should established its potential role in PASMC" – established to be changed to establish
3. Regarding this sentence: "Of the 29 tested associations in three TADs, the CpG marker (cg04917472) near the TSS of CTSZ showed the most marked associations with CTSZ itself" – it should be re-written to clarify that it is referring specifically to the cg04917472 probe, because of the 29 tested associations, the most marked association is with cg27396197 (MRPS31) based on p-values

in Supplementary Table 6.

4. In the methods, the information of TADs should be separate (as opposed to in the Proteomics section)
5. Re-check the order of Supplementary Tables – i.e., supplementary Table 2 is at the end (methods)

Reviewer #2 (Remarks to the Author):

Ulrich, Wu and colleagues performed an epigenome-wide association study of pulmonary arterial hypertension and followed up one of the most promising signals, at the CTSZ locus, with external datasets as well as experimental validations. I really like that this story takes a large-scale approach and continues all the way to target validation. I do have a number of questions and comments, outlined below. Moreover, the manuscript reads almost as two separate studies. The story could become much stronger if these two parts, the EWAS and the experiments, would be connected more strongly.

Main points

- It is understandable that the EWAS is done in the more easily accessible tissue of blood, rather than in the affected lung tissue. It would be good, though, to include an estimate of the correlation of methylation patterns in blood vs. pulmonary vascular cells.
- Since the EWAS results are quite modest, could you leverage a method like PascalX to identify gene p-values, rather than CpG-specific p-values?
- The analysis of the gene expression levels is a bit confusing. In line 175, the authors state: 'Of the tested 29 associations' Are these 29 genes that you tested? Or 29 CpGs with all genes in those regions? If the latter, how many genes were tested? And was the gene expression data from patients or healthy controls or both? It would be interesting to know if the association between cg04917472 and CTSZ expression is present only in patients or whether there are differences between patients and controls.
- Regarding the other 2 signals. Supp Fig 3 & 4 show that these CpG are the only ones associated in this region, with no typical peak of surrounding associations (like you see in Fig 3). Can the authors explain why these CpGs are so isolated? Is it possible that these are spurious associations? It would be good to also check gene expression & protein levels for these two other signals.
- One point that requires more clarification is the direction of effect of cathepsin. Do you have an idea why more methylated CTSZ associates with lower expression levels of CTSZ in patients, but patients also have a higher level of the protein? Lines 217-219 are unclear (what does "latterly inflammation" mean?). Could it be that what you measure in plasma is cathepsin that is excreted from cells or that ends up in plasma after cell death, and that the level is actually lower within cells of patients? It would be very helpful to make a cartoon representation of the biological mechanism as you understand/hypothesize the levels of CTSZ change along disease progression. In line with this point: Line 205: "but the direction of change suggested that white blood cells were not the primary determinant of systemically circulating protein." Please be explicit why this was suggested by the direction of effect.
- Methods: you corrected for 10 PCs computed from control probes and 5 "epi-structure" PCs. Please give more details on how those latter 5 were calculated, and why you correct for 5 (why not 10? Or 20?). Were the 10 PCs calculated and corrected for before correcting for the other 5? Or simultaneously? If simultaneously, do they really capture independent variation?
- Are the full summary statistics available?

Small comments

Abstract

- line 44: "At established PAH genes, .." ♦ Would be good to include here how many established PAH genes there are, and how many showed hypermethylation

Introduction

- Line 67: "new affected" ♦ "newly affected"

- Introduction: how many PAH cases are familial vs. idiopathic?
- Regarding the discrepancy between sexes in penetrance: is BMRP2 located on chr. X?
- Please briefly explain the difference between PAH and PH
- Line 110: "white cells" ◊ "white blood cells"

Results

- Did you use the same Illumina Infinium MethylationEPIC kit to profile all additional healthy controls? If so, I would start with the full number of controls immediately at line 121 and then describe the datasets that make up the total number afterwards.
- Figure 1: why are the number of CpGs different for the cohorts and going down in the pipeline when the exclusion criteria are only based on samples?
- Figure 2: please add Supp Fig 2 (lower) to this figure. I think it's important to see the raw data for these three loci.
- Figure 3: Would be clearer to order the tracks such that the same histone marks are underneath each other. And to add on the side the relevant information by a colored bar for donor ID, tissue and histone mark. Also add a color scale for the correlation between the lead CpG and the other CpGs in the region.
- Lines 160-164: Unclear. Did you take a window around the 16 genes and check methylation? Or did you somehow make a gene-level methylation score for these 16 genes? And then you found one marker that was hypermethylated? How many did you check?
- Supp Fig 12 and 13: For completeness, please also include Pcontrol vs IPAH
- Line 198: "We found that DNA methyltransferase DNMT3A and histone deacetylases HDAC1/2/6/11 were significantly different between PAH and controls" ◊ What was different? Expression levels? Or the association between CpG status and expression level?
- Fig 5
 - o A: please include the full gel picture, could be a supplement.
 - o A: Mention the reason for including Vinculin in the caption.
 - o Label each panel and order from A – F for clarity
- Line 230: Why did you choose TNF-a and LPS stimulation? Please expand this section a bit more, especially with the rationale of why you chose to look at certain aspects of biology in response to CTSZ downregulation.
- Line 235 "MTT assay" what's this abbreviation?

REVIEWER COMMENTS

Reviewer #1 (Remarks to the Author):

In the manuscript “Blood DNA Methylation Profiling Identifies Cathepsin Z Dysregulation in Pulmonary Arterial Hypertension” by Ulrich and colleagues, the authors tested whether blood-based methylation changes were associated with the rare pulmonary arterial hypertension (PAH). In order to test this, the authors performed an EWAS using the 850K methylationEPIC arrays in a cohort of 429 individuals with PAH and > 1,206 controls, and found that the cg04917472 probe, located upstream of the CTSZ gene was the most significant, and correlated with a decreased expression of the gene. Further functional validation revealed that the knockdown of the gene induced apoptosis, a hallmark of PAH.

General comments: This work is indeed novel, interesting, well-written and includes functional data that brings biological insight from the EWA study. There are some comments that need to be addressed, detailed below:

1. For the statistical analysis, the authors have tested > 865,000 CpG probes, however, there are probes that should be removed prior to the EWAS analysis, such as: cross-hybridizing probes, non-CpG probes, and probes near SNPs, that could confound CpG detection

a. For instance, based on supplementary Table 4, there is a SNP (rs114059951) located within one base pair of the cg04917472 CpG. Similarly for the other two CpG sites.

Most of these variants are very low frequency, for example in The Genome Aggregation Database (gnomAD, <https://gnomad.broadinstitute.org/>) rs114059951 is reported with a minor allele frequency - MAF of 0.003535 and therefore is going to be present in a very few individuals preventing it to drive any association and would not materially change our findings.

Indeed, the Lehne study which presented the pipeline investigated this further stating: “We found very little evidence to suggest these markers reduce overall data quality. Including them during quantile normalisation does not materially affect correlation between technical duplicates (mean $r = 0.9979$ in both cases). P value distributions under no association show no evidence that non-CpG markers or markers with SNPs in the probe sequence are more likely to generate spurious results, but we observe a slight increase in correlation for cross-hybridising markers (Additional file 1: Figure S25). We therefore recommend retaining, but flagging these markers.”

2. The general accepted threshold for the detection p-values is ≥ 0.01 , whereas the authors used 0.049– is there a reason for this?

This p-value threshold was again derived from the published Lehne pipeline study where they indicate it is based on Illumina recommendation. We have now added a detection p value to the **Supplementary Table 3** which shows that none of the CpGs with even a suggestive significance in this study had a p-value greater than 0.01, so changing this threshold would not have altered the study results.

3. What is the effect? Are these beta values? Would be good to indicate the % methylation in the text for the top CpG sites - for biological insight.

In the results text we present the odds ratios e.g. OR[95%CI]= 1.495[1.325-1.687] per % increase in methylation – we have clarified this by adding ‘per % increase in methylation’ to the first instance in the results section of the main text. We have also added raw betas into the main results text for biological reference. In the supplementary table we present effect of two different types of analyses and these are clarified as follows in the ‘legend’ tab on the excel file:

Effect¹	Effect estimate of CpG marker on PAH susceptibility in log(OR) per % increase in methylation
Effect²	Effect estimate of the CpG marker on transcript abundance (RNAseq) in beta units per TPM

4. It would be good to highlight the results from Supplementary Table 4: that there are several nominally significant CpGs in close proximity to the CTSZ probe, which are consistent in the direction of effect (i.e., hypermethylated)

We thank the reviewer for this point and have added a sentence as such to the results section of the main text.

5. In Figure 4A, the patients were divided into tertiles of CpG methylation (i.e., low, medium and high methylation) – it is not very clear why this was done? What (and why) is the a difference in analysis performed in Figure 4A vs. Supplementary Table 6?

This was simply done to illustrate the pattern of association more simply and clearly than the less visually appealing regression analysis (which keeps CpG methylation as a continuous variable) results presented in the table, but essentially, they show the same thing, i.e., that lower DNA methylation leads to higher RNA expression in this case.

6. In general, more detail should be included regarding the RNA expression vs. methylation in the methods section (in addition to the details in the figure legends)

We agree and have added the following explanation: “*Estimated gene abundances (in TPM) were then analysed in comparison to residuals of DNAm levels – betas adjusted for the EWAS covariates – for the lead CpGs against nearby genes within topologically associated domains (see below) correcting for multiple comparisons by FDR.*”

7. There is a difference in direction of effect between the RNA and protein, which the authors point out. However, this discordance was not fully explained or addressed. For instance, the authors mention: “It may be that in the early stages of disease, decreased CTSZ is associated with hypermethylation of the promoter, but latterly inflammation and renal dysfunction elevate CTSZ levels.”, but there is no evidence to support this.

Our main evidence to support this hypothesis is the analysis presented in **Supplementary Table 8** of plasma CTSZ protein levels and found they are positively associated with renal dysfunction (elevated cystatin C), inflammation (IL-6) but are actually negatively associated with cardiac biomarker NT-proBNP (which indicates the severity of the PH itself driving RV stress and failure). Unfortunately, due to the nature of the disease, patients present themselves for clinical diagnosis quite late in the disease pathogenesis, when the disease is well-established, and it is not possible to obtain clinical samples prior to significant vascular remodelling and inflammation occurring.

8. The other two probes that were significant also seem to be interesting – for instance the targeted gene *TUBB1* by the cg04917472 probe, in addition to the cg27396197 probe targeting *COG6* and *MRPS31* – was there a reason why these were not also further functionally studied? Or checked whether associated with protein levels?

The association with *CTSZ* was the strongest (highest magnitude of beta coefficient), which along with its interesting biology and the proximity of the CPGs to its promoter made it clearly the most promising priority for further study. The associations at *TUBB1* and *MRPS31* were also positive associations, which is less typical for DNA methylation associations (usually associated with gene silencing), but we present these results in the supplement such that the community are enabled to pursue these results further.

Minor comments:

1. The authors mention in the discussion: “it is important to note that while the EPIC array is the highest-coverage DNAm array platform available” – if referring to the 850K EPIC array, there is higher (Infinium MethylationEPIC v2.0 BeadChip). Therefore, could simply indicate that the 850K EPIC array only covers only < 4%.....

We have clarified, this was true only ‘at the time of analysis’.

2. In the sentence: “its effects on PAEC function but future efforts should established its potential role in PASMCM” – established to be changed to establish

We thank the reviewer for spotting this and have changed as suggested.

3. Regarding this sentence: “Of the 29 tested associations in three TADs, the CpG marker (cg04917472) near the TSS of CTSZ showed the most marked associations with CTSZ itself” – it should be re-written to clarify that it is referring specifically to the cg04917472 probe, because of the 29 tested associations, the most marked association is with cg27396197 (MRPS31) based on p-values in Supplementary Table 6.

We have clarified in the text this comment is based on it being the strongest effect (highest magnitude of beta coefficient) which we suggest is biologically more relevant than the similar p-values.

4. In the methods, the information of TADs should be separate (as opposed to in the Proteomics section)

We have changed this as suggested.

5. Re-check the order of Supplementary Tables – i.e., supplementary Table 2 is at the end (methods)

Done, thank you.

Reviewer #2 (Remarks to the Author):

Ulrich, Wu and colleagues performed an epigenome-wide association study of pulmonary arterial hypertension and followed up one of the most promising signals, at the CTSZ locus, with external datasets as well as experimental validations. I really like that this story takes a large-scale approach and continues all the way to target validation. I do have a number of questions and comments, outlined below. Moreover, the manuscript reads almost as two separate studies. The story could become much stronger if these two parts, the EWAS and the experiments, would be connected more strongly.

Main points

- It is understandable that the EWAS is done in the more easily accessible tissue of blood, rather than in the affected lung tissue. It would be good, though, to include an estimate of the correlation of methylation patterns in blood vs. pulmonary vascular cells.

This is an interesting point and will hopefully be better addressed in future studies. We have done this as far as it is currently possible to with the available data by validating our main associations in the publicly available dataset from pulmonary vascular endothelial cells, however this study is limited to a very small number of patients and therefore it is difficult to produce any further estimates of correlation beyond this. This is also affected by the differences in the assays performed – some of our lead CpGs were simply not available to check in that dataset thus limiting our ability to estimate the agreement between our data and the vascular cell findings.

- Since the EWAS results are quite modest, could you leverage a method like PascalX to identify gene p-values, rather than CpG-specific p-values?

The reviewer raises another interesting suggestion which will be an approach we will pursue in future work but is not within the scope of this study – we have added a statement to the limitations paragraph indicating as such – *“Future work will also pursue the analysis of DNA methylation on a regional or per-gene basis rather than at the single CpG level.”*

- The analysis of the gene expression levels is a bit confusing. In line 175, the authors state: ‘Of the tested 29 associations’ Are these 29 genes that you tested? Or 29 CpGs with all genes in those regions? If the latter, how many genes were tested? And was the gene expression data from patients or healthy controls or both? It would be interesting to know if the association between cg04917472 and CTSZ expression is present only in patients or whether there are differences between patients and controls.

We clarified in the results text that this is 29 genes but only tested once each *“(7-12 for each of three CpG/TADs)”*. We have further clarified that *‘Gene expression data were only available in PAH patients.’*

- Regarding the other 2 signals. Supp Fig 3 & 4 show that these CpG are the only ones associated in this region, with no typical peak of surrounding associations (like you see in Fig 3). Can the authors explain why these CpGs are so isolated? Is it possible that these are spurious associations? It would be good to also check gene expression & protein levels for these two other signals.

These 2 signals are within regions less densely represented on the array as shown by the fewer points on the **Supp Figs 3&4** and less correlation signified by the colouring of the points, so one would not have expected to see other associated CpGs here in the same way as observed in the CTSZ locus. We have analysed these two signals against RNA expression and cg27396197 showed a

positive correlation with *MRPS31* (Putative mitochondrial ribosomal protein S1, not expected to be measurable in plasma proteomics) but neither signal showed an expected negative correlation as observed with *CTSZ*.

- One point that requires more clarification is the direction of effect of cathepsin. Do you have an idea why more methylated *CTSZ* associates with lower expression levels of *CTSZ* in patients, but patients also have a higher level of the protein? Lines 217-219 are unclear (what does “latterly inflammation” mean?). Could it be that what you measure in plasma is cathepsin that is excreted from cells or that ends up in plasma after cell death, and that the level is actually lower within cells of patients? It would be very helpful to make a cartoon representation of the biological mechanism as you understand/hypothesize the levels of *CTSZ* change along disease progression. In line with this point: Line 205: “but the direction of change suggested that white blood cells were not the primary determinant of systemically circulating protein.” Please be explicit why this was suggested by the direction of effect.

We have clarified the use of latterly to ‘*later in the disease*’ in the text. We agree that the plasma cathepsin is likely secreted/released following cell death and that we would still predict cellular levels in white blood cells and the HPAEC of the published study would be lower, in line with the RNA data, but blood cell proteins from the same patients are not available for analysis. As such, we suggest a cartoon of the potential mechanism may be too speculative at this stage. We have also softened the highlighted sentence to say that white cells ‘*might not be*’ the primary source, clarifying the overall message as follows: “*the direction of change suggested that white blood cells (which we predict would contain lower *CTSZ* protein following reduced mRNA) might not be the primary determinant of (elevated) systemically circulating protein*”.

- Methods: you corrected for 10 PCs computed from control probes and 5 “epi-structure” PCs. Please give more details on how those latter 5 were calculated, and why you correct for 5 (why not 10? Or 20?). Were the 10 PCs calculated and corrected for before correcting for the other 5? Or simultaneously? If simultaneously, do they really capture independent variation?

Yes, first the control-probe PC outliers were removed, then the epi-structure PCs were calculated. The flow diagram in **Figure 1** presents the sequence of steps where “epi-structure PC” calculation is shown as the last step. We tested the use of 10 or 20 epi-structure PCs, and those gave no further improvement in the genomic inflation factor. The EPISTRUCTURE method, we used for calculating “epi-structure PCs”, is designed to capture ancestry-related variation (from Methods: “*Ancestry-related principal components were calculated with the EPISTRUCTURE method²⁰ implemented in GLINT software²¹*”), whereas the control probes are designed to capture information related to sample quality and thus should capture independent variation.

- Are the full summary statistics available?

Upon acceptance of the paper, we will upload the summary statistics to the www.ewascatalog.org website.

Small comments

Abstract

- line 44: "At established PAH genes, .." ☞ Would be good to include here how many established PAH genes there are, and how many showed hypermethylation

We clarified this as suggested (16 genes, only *BMP10* was significant).

Introduction

- Line 67: "new affected" ☞ "newly affected"

Corrected, thank you.

- Introduction: how many PAH cases are familial vs. idiopathic?

We clarified '*hereditary PAH (~6% of PAH cases)*,'.

- Regarding the discrepancy between sexes in penetrance: is *BMPR2* located on chr. X?

No, we have added that it is on chr2.

- Please briefly explain the difference between PAH and PH

On its first mention we have clarified PH '*can include pulmonary venous hypertension*' as well as PAH.

- Line 110: "white cells" ☞ "white blood cells"

Corrected as suggested.

Results

- Did you use the same Illumina Infinium MethylationEPIC kit to profile all additional healthy controls? If so, I would start with the full number of controls immediately at line 121 and then describe the datasets that make up the total number afterwards.

Yes, these were profiled on the same platform, therefore we have edited the text as suggested.

- Figure 1: why are the number of CpGs different for the cohorts and going down in the pipeline when the exclusion criteria are only based on samples?

As detailed in the methods *“For each batch (in case of the PAH Cohort Study) and dataset (in the case of external cohorts ADNI, PPMI and NFBC1966) separately, we removed CpG markers with detection p-value > 0.049 in over 50% of samples by setting their values to missing for all samples.”*. The final analysis was performed on all CpGs which met all criteria in all batches. For some studies, provided data had already been filtered using less stringent QC criteria, superseded by our own criteria in the final selection.

- Figure 2: please add Supp Fig 2 (lower) to this figure. I think it's important to see the raw data for these three loci.

We have moved these figures as suggested.

- Figure 3: Would be clearer to order the tracks such that the same histone marks are underneath each other. And to add on the side the relevant information by a colored bar for donor ID, tissue and histone mark. Also add a color scale for the correlation between the lead CpG and the other CpGs in the region.

We added the colour scale bar for the correlation of CpGs as suggested and side labels to detail the tissue and then histone mark more clearly, so it is easier for the reader to follow. We feel grouping the tracks by tissue is more intuitive and therefore have left it in this order, plus not all histone marks are available for all tissues which is why it is often not possible to plot as suggested.

- Lines 160-164: Unclear. Did you take a window around the 16 genes and check methylation? Or did you somehow make a gene-level methylation score for these 16 genes? And then you found one marker that was hypermethylated? How many did you check?

We analysed CpGs annotated by Illumina as being associated with the 16 genes individually. As detailed in the methods *“557 CpG markers at 16 established genes underlying heritable PAH*

catalogued by Southgate et al. [24] were assessed in a targeted sub-analysis reported with a FDR-corrected q threshold based on the number of markers tested."

- Supp Fig 12 and 13: For completeness, please also include Pcontrol vs IPAH

We added these p values as suggested to the supplemental figures.

- Line 198: "We found that DNA methyltransferase DNMT3A and histone deacetylases HDAC1/2/6/11 were significantly different between PAH and controls" □ What was different? Expression levels? Or the association between CpG status and expression level?

We have clarified this is expression levels.

- Fig 5

o A: please include the full gel picture, could be a supplement.

This is submitted as per Nature guidelines.

o A: Mention the reason for including Vinculin in the caption.

We have added: "*(compared to loading controls vinculin and beta-actin).*"

o Label each panel and order from A – F for clarity

We changed this figure as suggested and agree it became clearer.

- Line 230: Why did you choose TNF-a and LPS stimulation? Please expand this section a bit more, especially with the rationale of why you chose to look at certain aspects of biology in response to CTSZ downregulation.

We clarified this section in beginning '*In PAH, the pulmonary endothelium becomes pro-proliferative, anti-apoptotic and disorganised, partly in response to inflammatory stimuli*'. Hence, pro-inflammatory TNF-a or LPS stimulation partially mimics the disease process.

- Line 235 “MTT assay” what’s this abbreviation?

We clarified MTT is a tetrazolium dye assay, the principal being the metabolism of MTT reflects viable/metabolically active cell count (described in figure legend), and under pro-proliferative conditions (normal culture of ECs, especially in presence of VEGF is proliferative) this therefore represents proliferation.

REVIEWERS' COMMENTS

Reviewer #1 (Remarks to the Author):

In response to the Author's comments of the manuscript "Blood DNA Methylation Profiling Identifies Cathepsin Z Dysregulation in Pulmonary Arterial Hypertension" by Ulrich et al:

The authors have addressed most comments sufficiently. This work is noteworthy and the EWAS does indeed provide an interesting and significant results in the field.

- 1) The reviewer encourages the authors to incorporate their response to the comment regarding the SNP in the CpG site, specifically addressing the low frequency of the SNP, into the manuscript.
- 2) Presenting the raw beta values as percentages would be better.

Reviewer #2 (Remarks to the Author):

The authors have answered most of my points sufficiently. I would suggest for figure 2 to name each of the plots in the lower half of the figure by the region highlighted in the Manhattan plot in the upper half. I would also ask to include where people can find the summary statistics on ewascatalog.org, as the authors wrote they will deposit them there upon publication.

Reviewer #1 (Remarks to the Author):

In response to the Author's comments of the manuscript "Blood DNA Methylation Profiling Identifies Cathepsin Z Dysregulation in Pulmonary Arterial Hypertension" by Ulrich et al:

The authors have addressed most comments sufficiently. This work is noteworthy and the EWAS does indeed provide an interesting and significant results in the field.

We thank the reviewer for their comments.

- 1) The reviewer encourages the authors to incorporate their response to the comment regarding the SNP in the CpG site, specifically addressing the low frequency of the SNP, into the manuscript.

We have added the following sentence to the discussion 'There is a SNP (rs114059951) located within one base pair of the cg04917472 CpG; in The Genome Aggregation Database (gnomAD) rs114059951 is reported with a minor allele frequency - MAF of 0.003535 and therefore is going to be present in a very few individuals and would not materially change our findings.'

- 2) Presenting the raw beta values as percentages would be better.

We have made this change as suggested.

Reviewer #2 (Remarks to the Author):

The authors have answered most of my points sufficiently. I would suggest for figure 2 to name each of the plots in the lower half of the figure by the region highlighted in the Manhattan plot in the upper half.

We have made this change as suggested.

I would also ask to include where people can find the summary statistics on ewascatalog.org, as the authors wrote they will deposit them there upon publication.

We have deposited these and added 'The EWAS summary statistics will be deposited at ewascatalog.org accessible at DOI:10.5281/zenodo.10276821.' to the methods section.